# META-CHUNKING: LEARNING EFFICIENT TEXT SEGMENTATION VIA LOGICAL PERCEPTION

## ABSTRACT

Retrieval-Augmented Generation (RAG), while serving as a viable complement to large language models (LLMs), often overlooks the crucial aspect of text chunking within its pipeline, which impacts the quality of knowledge-intensive tasks. This paper introduces the concept of Meta-Chunking, which refers to a granularity between sentences and paragraphs, consisting of a collection of sentences within a paragraph that have deep linguistic logical connections. To implement Meta-Chunking, we designed Perplexity (PPL) Chunking, which balances performance and speed, and precisely identifies the boundaries of text chunks by analyzing the characteristics of context perplexity distribution. Additionally, considering the inherent complexity of different texts, we propose a strategy that combines PPL Chunking with dynamic merging to achieve a balance between fine-grained and coarse-grained text chunking. Experiments conducted on eleven datasets demonstrate that Meta-Chunking can more efficiently improve the performance of single-hop and multi-hop question answering based on RAG. For instance, on the 2WikiMultihopQA dataset, it outperforms similarity chunking by 1.32 while only consuming 45.8% of the time. Furthermore, through the analysis of models of various scales and types, we observed that PPL Chunking exhibits notable flexibility and adaptability.

## 1 INTRODUCTION

Retrieval-augmented generation (RAG), as a cutting-edge technological paradigm, aims to address challenges faced by large language models (LLMs), such as data freshness (He et al., 2022), hallucinations (Bénédict et al., 2023; Chen et al., 2023b; Zuccon et al., 2023; Liang et al., 2024), and the lack of domain-specific knowledge (Li et al., 2023; Shen et al., 2023). This is particularly relevant in knowledge-intensive tasks like open-domain question answering (Lazaridou et al., 2022). By integrating two key components: the retriever and the generator, this technology enables more precise responses to input queries (Singh et al., 2021; Lin et al., 2023). While the feasibility of the retrieval-augmentation strategy has been widely demonstrated through practice, its effectiveness heavily relies on the relevance and accuracy of the retrieved documents (Li et al., 2022; Tan et al., 2022). The introduction of excessive redundant or incomplete information through retrieval not only fails to enhance the performance of the generation model but may also lead to a decline in answer quality (Shi et al., 2023; Yan et al., 2024).

In response to the aforementioned challenges, current research efforts mainly focus on two aspects: improving retrieval accuracy (Zhuang et al., 2024; Sidiropoulos & Kanoulas, 2022; Guo et al., 2023) and enhancing the robustness of LLMs against toxic information (Longpre et al.; Kim et al., 2024). However, in RAG systems, a commonly overlooked aspect is the chunked processing of textual content, which directly impacts the quality of dense retrieval (Xu et al., 2023). By delicately splitting long documents into multiple chunks, this module not only significantly improves the processing efficiency and performance of the system, reducing the consumption of computing resources, but also enhances the accuracy of retrieval (Besta et al., 2024). Meanwhile, the chunking strategy allows information to be more concentrated, minimizing the interference of irrelevant information, enabling LLMs to focus more on the specific content of each text chunk and generate more precise responses (Su et al., 2024). Traditional text chunking methods, often based on rules or semantic similarity (Zhang et al., 2021; Langchain, 2023; Lyu et al., 2024), provide some structural segmentation but are inadequate in capturing subtle changes in logical relationships between sentences. As illustrated

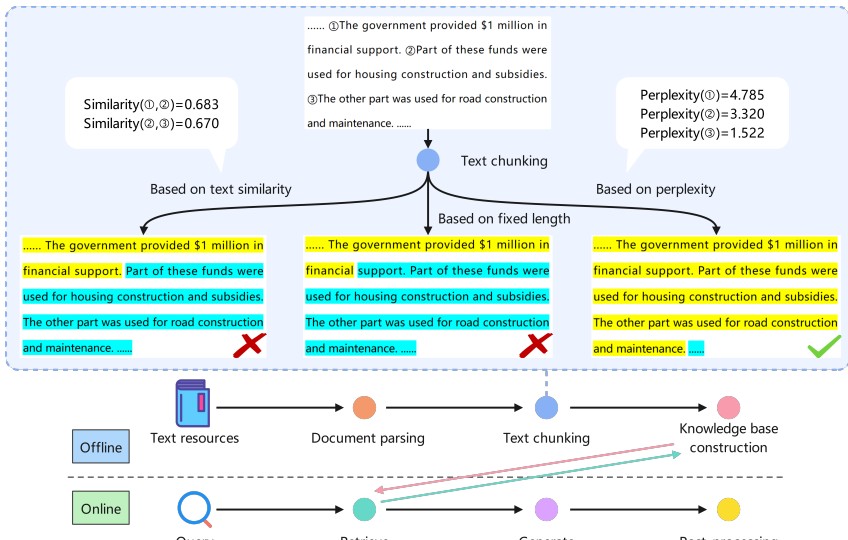

Figure 1: Overview of RAG pipeline, as well as examples based on rules, similarity, and PPL segmentation. The same background color represents being located in the same chunk.

in Figure 1, example sentences exhibit a progressive relationship, yet their semantic similarity is low, which may result in their complete separation. The LumberChunker (Duarte et al., 2024) offers a novel solution by utilizing LLMs to receive a series of consecutive paragraphs and accurately identify where content begins to diverge. However, it demands a high level of instruction-following ability from LLMs, necessitating the use of the Gemini model, which incurs significant resource and time costs. This raises a practical question: How can we fully utilize the powerful reasoning capabilities of LLMs while efficiently accomplishing the text chunking task at a lower cost?

This paper introduces the concept of **Meta-Chunking**, which operates at a granularity between sentences and paragraphs, aiming to enhance logical coherence in the process of text segmentation. Meta-Chunking consists of sets of sentences within paragraphs that share deep linguistic and logical connections. To address the limitations of traditional methods based on semantic similarity, we leverage the powerful comprehension and reasoning capabilities of LLMs to devise the Meta-Chunking strategy: **Perplexity (PPL) Chunking**. This method calculates the PPL of each sentence based on its context and identifies text chunk boundaries by analyzing the characteristics of PPL distribution. It effectively reduces the dependency of text chunking on model scale, enabling smaller language models with relatively weaker reasoning capabilities to adequately perform this task. Furthermore, PPL Chunking improves the efficiency of LLMs in handling chunking tasks, achieving both resource and time savings. This provides crucial support for LLMs to process text chunking in real-world scenarios.

To comprehensively evaluate proposed methods, extensive experiments were conducted on eleven datasets across four benchmarks, involving both Chinese and English texts, ranging from brief to extensive documents, and measured through seven key metrics. In response to the inherent complexity of different datasets, we propose a Meta-Chunking with dynamic combination strategy designed to achieve a valid balance between fine-grained and coarse-grained text segmentation. Traditional chunking methods treat sentences as independent logical units, whereas we adopt meta-chunks as independent logical units. For instance, in the RAG system, if users opt for a small model and set a relatively low top_k value for recall, meta-chunks can be directly utilized. However, in cases where users employ LLMs with extended contexts and require larger text chunks, meta-chunks can initially be generated and subsequently merged based on the desired chunk size to achieve the final chunking outcome. Experimental results fully demonstrate that the Meta-Chunking strategy significantly improves performance compared to traditional rule-based and semantic chunking. More importantly, compared to the current LLMs approach, the method proposed in this paper exhibits superior performance in terms of efficiency and cost savings.

## 2 RELATED WORKS

**Text Segmentation** It is a fundamental task in NLP, aimed at breaking down text content into its constituent parts to lay the foundation for subsequent advanced tasks such as information retrieval (Li et al., 2020) and text summarization (Lukasik et al., 2020; Cho et al., 2022). By conducting topic modeling on documents, Kherwa & Bansal (2020) and Barde & Bainwad (2017) demonstrate the identification of primary and sub-topics within documents as a significant basis for text segmentation. Numerous techniques exist for topic modeling, ranging from algorithms based on probabilistic methods, such as Latent Dirichlet Allocation (Blei et al., 2003) and Probabilistic Latent Semantic Analysis (Hofmann et al., 1999), to models that also consider semantic relationships between words and sentences, like Top2Vec (Angelov, 2020) and BERTopic (Grootendorst, 2022). Additionally, Zhang et al. (2021) frames text segmentation as a sentence-level sequence labeling task, utilizing BERT to encode multiple sentences simultaneously. It calculates sentence vectors after modeling longer contextual dependencies and finally predicts whether to perform text segmentation after each sentence. Langchain (2023) provides flexible and powerful support for various text processing scenarios by integrating multiple text segmentation methods, including character segmentation, delimiter-based text segmentation, specific document segmentation, and recursive chunk segmentation. Although these methods better respect the structure of the document, they have limitations in deep contextual understanding. To address this issue, semantic-based segmentation (Kamradt, 2024) utilizes embeddings to aggregate semantically similar text chunks and identifies segmentation points by monitoring significant changes in embedding distances.

**Text Chunking in RAG** LLMs have demonstrated remarkable capabilities in language-related tasks through their complex internal structures and reasoning mechanisms (Zheng et al., 2024). By expanding the input space of LLMs through introducing retrieved text chunks (Guu et al., 2020; Lewis et al., 2020), RAG significantly improves the performance of knowledge-intensive tasks (Ram et al., 2023). Text chunking plays a crucial role in RAG, as ineffective chunking strategies can lead to incomplete contexts or excessive irrelevant information, thereby hurting the performance of QA systems (Yu et al., 2023). Besides typical granularity levels like sentences or paragraphs (Lyu et al., 2024; Gao et al., 2023), there are other advanced methods available. Chen et al. (2023a) introduced a novel retrieval granularity called Proposition, which is the smallest text unit that conveys a single fact. This method excels in fact-based texts like Wikipedia. However, it may not perform ideally when dealing with content that relies on flow and contextual continuity, such as narrative texts, leading to the loss of critical information. Meanwhile, LumberChunker (Duarte et al., 2024) iteratively harnesses LLMs to identify potential segmentation points within a continuous sequence of textual content, showing some potential for LLMs chunking. However, this method demands a profound capability of LLMs to follow instructions and entails substantial consumption when employing the Gemini model.

## 3 METHODOLOGY

### 3.1 META-CHUNKING

Our main contribution is an innovative text segmentation technique named **Meta-Chunking**, which leverages the capabilities of LLMs to flexibly partition documents into logically coherent, independent chunks. Our approach is grounded in a core principle: allowing variability in chunk size to more effectively capture and maintain the logical integrity of content. This dynamic adjustment of granularity ensures that each segmented chunk contains a complete and independent expression of ideas, thereby avoiding breaks in the logical chain during the segmentation process. This not only enhances the relevance of document retrieval but also improves content clarity.

As illustrated in Figure 2, our method integrates the advantages of traditional text segmentation strategies, such as adhering to preset chunk length constraints and ensuring sentence structural integrity, while enhancing the ability to guarantee logical coherence during the segmentation process. The key lies in introducing a novel concept between sentence-level and paragraph-level text granularity: **Meta-Chunking**. A meta chunk consists of a collection of sequentially arranged sentences within a paragraph, where the sentences not only share semantic relevance but, more importantly, contain deep linguistic logical connections, including but not limited to causal, transitional, parallel,

and progressive relationships. These relationships go beyond mere semantic similarity. In order to achieve this goal, we have designed and implemented the following strategy.

**Perplexity Chunking**: Given a text, the initial step involves segmenting it into a collection of sentences denoted as $(x_1, x_2, \ldots, x_n)$, with the ultimate goal being to further partition these sentences into several chunks, forming a new set $(X_1, X_2, \ldots, X_k)$, where each chunk comprises a coherent grouping of the original sentences. We split the text into sentences and use the model to calculate the PPL of each sentence $x_i$ based on the preceding sentences:

$$\text{PPL}_M(x_i) = \frac{\sum_{k=1}^{K} \text{PPL}_M(t_k^i | t_{<k}^i, t_{<i})}{K} \tag{1}$$

where $K$ represents the total number of tokens in $x_i$, $t_k^i$ denotes the $k$-th token in $x_i$, and $t_{<i}$ signifies all tokens that precede $x_i$. To locate the key points of text segmentation, the algorithm further analyzes the distribution characteristics of $\text{PPL}_{seq} = (\text{PPL}_M(x_1), \text{PPL}_M(x_2), \ldots, \text{PPL}_M(x_n))$, particularly focusing on identifying minima:

$$\text{Minima}_{index}(\text{PPL}_{seq}) = \left\{ i \; \middle| \; \min(\text{PPL}_M(x_{i-1}), \text{PPL}_M(x_{i+1})) - \text{PPL}_M(x_i) > \theta, \right.$$

$$\left. \text{or } \text{PPL}_M(x_{i-1}) - \text{PPL}_M(x_i) > \theta \text{ and } \text{PPL}_M(x_{i+1}) = \text{PPL}_M(x_i) \right\} \tag{2}$$

The meaning of the above formula include: when the PPL on both sides of a point are higher than at that point, and the difference on at least one side exceeds the preset threshold $\theta$; or when the difference between the left point and the point is greater than $\theta$ and the right point equals the point value. These minima are regarded as potential chunk boundaries. If the text exceeds the processing range of LLMs or device, we strategically introduce a key-value (KV) caching mechanism. Specifically, the text is first divided into several parts according to tokens, forming multiple subsequences. As the PPL calculation progresses, when the GPU memory is about to exceed the server configuration or the maximum context length of LLMs, the algorithm appropriately removes KV pairs of previous partial text, thus not sacrificing too much contextual coherence.

To address diverse chunking needs of users, merely adjusting the threshold to control chunk size sometimes leads to uneven chunking sizes as the threshold increases, as shown in Section 5.2.2 and 5.2.3. Therefore, we propose a strategy combining Meta-Chunking with dynamic merging, aiming to flexibly respond to varied chunking requirements. Firstly, we set an initial threshold of 0 or a specific value based on the PPL distribution and perform Meta-Chunking operations, preliminarily dividing the document into a series of basic units $(c_1, c_2, \ldots, c_\alpha)$. Subsequently, according to the user-specified chunk length $L$, we iteratively merge adjacent meta-chunks until the total length satisfies or approximates the requirement. Specifically, if $\text{len}(c_1, c_2, c_3) = L$ or $\text{len}(c_1, c_2, c_3) < L$ while $\text{len}(c_1, c_2, c_3, c_4) > L$, then $c_1, c_2, c_3$ are regarded as a complete chunk.

## 3.2 THEORETICAL ANALYSIS OF PPL CHUNKING

LLMs are designed to learn a distribution $Q$ that approximates the empirical distribution $P$ from sample texts. To quantify the closeness between these two distributions, cross-entropy is typically employed as a metric. Under the discrete scenario, cross-entropy of $Q$ relative to $P$ is formally defined as follows:

$$H(P, Q) = \text{E}_p[-logQ] = -\sum_x P(x) \log Q(x) = H(P) + D_{KL}(P||Q) \tag{3}$$

where $H(P)$ represents the empirical entropy, and $D_{KL}(P||Q)$ is the Kullback-Leibler (KL) divergence between $Q$ and $P$. The PPL of LLMs, mathematically speaking, is defined as:

$$\text{PPL}(P, Q) = 2^{H(P,Q)} \tag{4}$$

It is essential to notice that, since $H(p)$ is unoptimizable and bounded as shown in Appendix A.1, what truly impacts the discrepancy in PPL calculations across different LLMs is the KL divergence, which serves as a metric to assess the difference between distributions. The greater the KL divergence is, the larger the disparity between two distributions signifies. Furthermore, high PPL

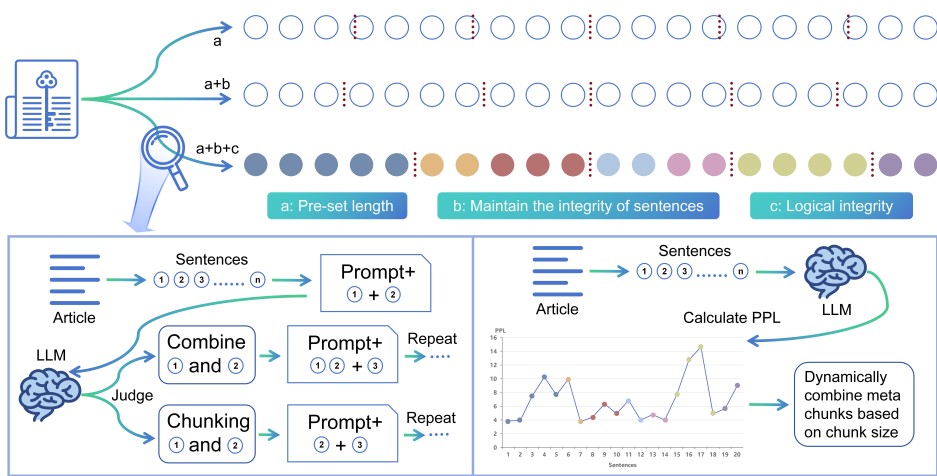

Figure 2: Overview of the entire process of Meta-Chunking. Each circle represents a complete sentence, and the sentence lengths are not consistent. The vertical lines indicate where to segment. The two sides at the bottom of the figure reveal Margin Sampling Chunking and Perplexity Chunking. Circles with the same background color represent a meta-chunk, which is dynamically combined to make the final chunk length meet user needs.

indicates the cognitive hallucination of LLMs towards the real content, and such portions should not be segmented.

On the other hand, Shannon (1951) approximates the entropy of any language through a function

$$G_K = - \sum_{T_k} P(T_k) \log_2 P(t_k|T_{k-1})$$

$$= - \sum_{T_k} P(T_k) \log_2 P(T_k) + \sum_{T_{k-1}} P(T_{k-1}) \log_2 P(T_{k-1}) \qquad (5)$$

where $T_k$ represents $k$ consecutive tokens $(t_1, t_2, \ldots, t_k)$ in a text sequence, entropy can then be expressed as

$$H(P) = \lim_{K \to \infty} G_K \qquad (6)$$

Then, based on the proof in Appendix A.1 that $G_{K+1} \leq G_K$ for all $K \geq 1$, we can derive

$$G_1 \geq G_2 \geq \cdots \geq \lim_{K \to \infty} G_K = H(P) \qquad (7)$$

By combining formulas (3) and (7), we observe that for large-scale text processing tasks, increasing the context length tends to reduce the cross-entropy or PPL, a phenomenon that reflects the ability of LLMs to make more effective logical inferences and semantic understandings after capturing broader contextual information. Consequently, during PPL Chunking experiments, we maximize the input of longer text sequences to LLMs, anticipating more substantial performance gains.

## 4 EXPERIMENT

### 4.1 DATASETS AND METRICS

We conducted a comprehensive evaluation on four benchmarks and comparison between Meta-Chunking and multiple baselines on a series of question answering (QA) datasets, focusing on both Chinese and English languages, and covering multiple metrics such as the correctness of answers, factuality, and recall of retrieved texts. The CRUD benchmark (Lyu et al., 2024) is a Chinese dataset containing single-hop, two-hop, and three-hop questions, evaluated using metrics including BLEU series, ROUGE-L, and BERTScore. We utilize the CUAD dataset from RAGBench benchmark (Friel et al., 2024), employing the same evaluation metrics as the CRUD. The MultiHop-RAG

benchmark (Tang & Yang) assesses recall rates, with metrics such as Hits@ series, MAP@10, and MRR@10. LongBench benchmark (Bai et al., 2023) comprises various datasets, among which we exploit eight Chinese and English datasets covering single and multi-hop QA, evaluated based on F1 and ROUGE-L metrics.

## 4.2 BASELINES

We primarily compared Meta-Chunking with two types of methods, namely rule-based chunking and dynamic chunking, noting that the latter incorporates both semantic similarity models and LLMs. The original rule-based method simply divides long texts into fixed-length chunks, disregarding sentence boundaries. However, the Llama_index method (Langchain, 2023) offers a more nuanced approach, balancing the maintenance of sentence boundaries while ensuring that token counts in each segment are close to a preset threshold. On the other hand, similarity chunking (Xiao et al., 2023) utilizes sentence embedding models to segment text based on semantic similarity, effectively grouping highly related sentences together. Dense X Retrieval (Chen et al., 2023a) introduces a new retrieval granularity called propositions, which condenses and segments text by training an information extraction model. Alternatively, LumberChunker (Duarte et al., 2024) employs LLMs to predict optimal segmentation points within the text. These methods exhibit unique strengths in adapting to the context and structure of texts.

It is noteworthy that LumberChunker encounters difficulties when applied to smaller models, thus impeding the comparison among different methods within the same model. To address this limitation, we introduced a Margin Sampling (MSP) strategy to optimize the method, enhancing its adaptability to smaller models. This optimization enables a more effective comparison of the performance and time consumption of various chunking methods.

**Margin Sampling Chunking**: We split the text into a collection of sentences denoted as $(x_1, x_2, \ldots, x_n)$, and the method can be formulated as:

$$\text{Margin}_M(x_i) = P_M\left(y = k_1 | \text{Prompt}(x_i, X^{'})\right) - P_M\left(y = k_2 | \text{Prompt}(x_i, X^{'})\right) \tag{8}$$

where $(k_1, k_2)$ indicates a binary decision between $yes$ or $no$ for a segmentation judgment. $\text{Prompt}(x_i, X^{'})$ represents forming an instruction between $x_i \in \{x_l\}_{l=1}^n$ and $X^{'}$, regarding whether they should be merged, where $X^{'}$ encompasses either a single sentence or multiple sentences. Through the probability $P_M$ obtained by model $M$, we can derive the probability difference $\text{Margin}_M(x_i)$ between the two options. Subsequently, by contrasting $\text{Margin}_M(x_i)$ with the threshold $\theta$, a conclusion can be drawn regarding whether the two sentences should be segmented. For the setting of $\theta$, we initially assign it a value of 0 and then adjust it by recording historical $\text{Margin}_M(x_i)$ and calculating their average.

## 4.3 EXPERIMENTAL SETTINGS

We primarily use Qwen2-0.5B, Qwen2-1.5B, Qwen2-7B and Baichuan2-7B for Meta-Chunking (Yang et al., 2024; 2023). Without additional annotations, all language models used in this paper adopt chat or instruction versions. When chunking, the default parameter configurations of the models are adopted. For evaluation, Qwen2-7B is employed with the following settings: top_p = 0.9, top_k = 5, temperature = 0.1, and max_new_tokens = 1280. When conducting QA, the system necessitates dense retrievals from the vector database, with top_k set to 8 for CRUD and RAGBench, 10 for MultiHop-RAG, and 5 for LongBench. Text segmentation in the dataset is performed using NVIDIA H800, and evaluation is conducted using NVIDIA GeForce RTX 3090. To control variables, we maintain consistent chunk lengths for various chunking methods across each dataset. Detailed experimental setup information can be found in Appendix A.2.

# 5 RESULTS AND ANALYSIS

## 5.1 MAIN RESULTS

**Comparison against Baselines.** We systematically evaluated the performance of five baseline methods, as shown in Table 1 (top) and Table 2 (top). Notably, LumberChunker with Qwen2-7B

Table 1: Main experimental results are presented in five QA datasets. The first four datasets are sourced from LongBench. Besides Dense X Retrieval, we maintain a consistent chunk length for various chunking methods in each dataset.

| Dataset | 2WikiMultihopQA | | Qasper | | MultiFieldQA-en | | MultiFieldQA-zh | | MultiHop-RAG | | | |
| Chunking Method | F1 | Time | F1 | Time | F1 | Time | F1 | Time | Hits@10 | Hits@4 | MAP@10 | MRR@10 |
|---|---|---|---|---|---|---|---|---|---|---|---|---|
| *Baselines with rule-based or similarity-based chunking* | | | | | | | | | | | | |
| Original | 11.89 | 0.21 | 9.45 | 0.13 | 29.89 | 0.16 | 22.45 | 0.06 | 0.6027 | 0.4523 | 0.1512 | 0.3507 |
| Llama_index | 11.74 | 8.12 | 10.15 | 5.81 | 28.30 | 6.25 | 21.85 | 5.53 | 0.7366 | 0.5437 | 0.1889 | 0.4068 |
| Similarity Chunking | 12.00 | 416.45 | 9.93 | 307.05 | 29.19 | 318.41 | 22.39 | 134.80 | 0.7232 | 0.5362 | 0.1841 | 0.3934 |
| Dense X Retrieval | 5.49 | 57633.07 | 8.23 | 39762.54 | 29.72 | 41789.49 | - | - | - | - | - | - |
| *Chunking based on Qwen2-0.5B* | | | | | | | | | | | | |
| MSP Chunking | 11.74 | 788.30 | **9.67** | 599.97 | **31.28** | 648.76 | 23.35 | 480.35 | 0.7162 | 0.5246 | 0.1830 | 0.3913 |
| PPL Chunking | **13.56** | **140.54** | 9.62 | **65.45** | 31.02 | **79.72** | **23.52** | **64.02** | **0.7215** | **0.5583** | **0.1925** | **0.4186** |
| *Chunking based on Qwen2-1.5B* | | | | | | | | | | | | |
| MSP Chunking | 11.30 | 2189.29 | 9.49 | 1487.27 | **32.81** | 1614.01 | 22.08 | 1881.15 | 0.7109 | 0.5517 | 0.1970 | 0.4252 |
| PPL Chunking | **13.32** | **190.93** | **9.82** | **122.44** | 31.30 | **136.96** | **22.57** | **107.94** | **0.7366** | **0.5570** | **0.1979** | **0.4300** |

achieved a score of 10.65 and a chunking time of 2883.43 seconds on the Qasper dataset but failed to work effectively on the other four datasets. This indicates significant limitations of this strategy in adapting to models with 7B parameters and below. Dense X Retrieval condenses and segments text by training an information extraction model, which does not allow for specifying the chunk length. Aside from this method, we maintain a consistent chunk length for various other chunking approaches across each dataset, which are enumerated individually in Appendix A.2.

As shown in Table 1 (bottom), PPL Chunking provides notable improvements in the performance of QA systems and information retrieval when utilizing models with 0.5B and 1.5B parameter scales. Specifically, both model configurations show measurable improvements in accuracy and recall metrics compared to baseline tasks. Furthermore, they exhibit significant enhancements in processing speed when compared to dynamic chunking, thereby facilitating easier implementation of LLMs chunking in real-world scenarios.

**Efficiency and Accuracy Trade-off.** Margin Sampling Chunking addresses the current issue where LLMs chunking cannot be applied to models with weak instruction-following capabilities, and it demonstrates superior performance compared to LumberChunker, as illustrated in Table 2. However, this method exhibits chunking times similar to the LumberChunker algorithm, both reaching threshold ranges that are challenging for practical applications, highlighting inefficiencies of LLMs in handling chunking tasks. In contrast, PPL Chunking demon-

Table 2: Main experimental results of LLMs chunking using Qwen2-7B. Consistent chunk lengths were maintained for various chunking methods in each dataset. *base* represents the basic model, while *inst.* denotes the model fine-tuned with instructions.

| Dataset | 2WikiMultihopQA | | Qasper | | MultiFieldQA-en | |
| Chunking Method | F1 | Time | F1 | Time | F1 | Time |
|---|---|---|---|---|---|---|
| Similarity Chunking | 12.00 | 416.45 | 9.93 | 307.05 | 29.19 | 318.41 |
| LumberChunker$_{inst.}$ | - | - | 10.65 | 2883.43 | - | - |
| MSP Chunking$_{inst.}$ | 12.94 | 8781.82 | **11.37** | 5755.79 | **33.56** | 6287.31 |
| PPL Chunking$_{base}$ | **14.15** | 745.11 | 10.11 | 493.43 | 30.92 | 530.22 |
| PPL Chunking$_{inst.}$ | 13.41 | **736.69** | 9.39 | 486.48 | 32.35 | **523.74** |

strates significant advantages, not only excelling in maintaining or approaching the performance level provided by Margin Sampling Chunking, but also achieving a substantial leap in processing efficiency compared to dynamic chunking strategies. Upon deeper examination between the base model and the instruction model, we found that PPL Chunking exhibits a remarkable flexibility and adaptability, indicating that it does not have a stringent requirement for the capacity of model to follow specific instructions.

**How Weak Can the Weaker LLM Be?** As a fundamental task, text chunking consumes a large number of tokens when using LLMs like GPT-4 or Gemini, often leading to a significant imbalance between resource utilization and task benefits. Therefore, using a lightweight model is a practical choice. Since our method is applicable to both large and small models, in addition to testing 1.5B

and 7B models, we explored smaller models below 1B parameters. As the model size decreases, the execution time of the text chunking task significantly reduces, reflecting the advantage of small models in improving processing efficiency. Furthermore, our approaches do not suffer from significant performance degradation as the model size decreases, and it outperforms baselines on most datasets, which further demonstrates the superiority of our methods.

## 5.2 ANALYSIS

### 5.2.1 IMPACT OF OVERLAPPING CHUNKING STRATEGIES

As we delve deeper into the influence of text chunking strategies on the performance of complex QA tasks, we further investigated the performance of various chunking strategies when overlapping chunks were employed. The original chunking overlap method uses a fixed number of characters from the end of one chunk to overlap with the start of the next. The Llama_index overlap approach builds upon this by additionally considering sentence integrity. The PPL Chunking overlap strategy, on the other hand, dynamically assigns sentences represented by minimal points of PPL to both the preceding and subsequent chunks, resulting in dynamic overlap. These approaches generally produce overlap lengths averaging around 50 Chinese characters.

Table 3: Performance of different methods on CRUD QA datasets with overlapping chunks. *ppl* represents direct PPL Chunking, with a threshold of 0.5. Precise chunk length and overlap length results are included in Appendix A.3.

| Chunking Method | Overlap | BLEU-1 | BLEU-2 | BLEU-3 | BLEU-4 | BLEU-Avg | ROUGE-L | BERTScore |
|---|---|---|---|---|---|---|---|---|
| *Single-hop Query* | | | | | | | | |
| Original | Fixed | 0.3330 | 0.2641 | 0.2214 | 0.1881 | 0.2410 | 0.4060 | 0.8425 |
| Llama_index | Dynamic | 0.3326 | 0.2645 | 0.2214 | 0.1890 | 0.2413 | 0.4039 | 0.8439 |
| Qwen2-1.5B$_{ppl}$ | Dynamic | 0.3592 | 0.2888 | 0.2435 | 0.2081 | 0.2644 | 0.4332 | 0.8555 |
| Qwen2-7B$_{ppl}$ | Dynamic | 0.3582 | 0.2898 | 0.2450 | 0.2097 | 0.2657 | 0.4308 | 0.8548 |
| Baichuan2-7B$_{ppl}$ | Dynamic | 0.3656 | 0.2952 | 0.2497 | 0.2143 | 0.2705 | 0.4393 | 0.8549 |
| *Two-hop Query* | | | | | | | | |
| Original | Fixed | 0.2251 | 0.1300 | 0.0909 | 0.0689 | 0.1114 | 0.2579 | 0.8747 |
| Llama_index | Dynamic | 0.2223 | 0.1282 | 0.0896 | 0.0677 | 0.1099 | 0.2555 | 0.8732 |
| Qwen2-1.5B$_{ppl}$ | Dynamic | 0.2295 | 0.1331 | 0.0934 | 0.0709 | 0.1143 | 0.2609 | 0.8700 |
| Qwen2-7B$_{ppl}$ | Dynamic | 0.2312 | 0.1353 | 0.0949 | 0.0719 | 0.1162 | 0.2638 | 0.8751 |
| Baichuan2-7B$_{ppl}$ | Dynamic | 0.2336 | 0.1350 | 0.0940 | 0.0710 | 0.1154 | 0.2650 | 0.8754 |
| *Three-hop Query* | | | | | | | | |
| Original | Fixed | 0.2384 | 0.1268 | 0.0832 | 0.0602 | 0.1066 | 0.2546 | 0.8823 |
| Llama_index | Dynamic | 0.2331 | 0.1250 | 0.0825 | 0.0598 | 0.1049 | 0.2517 | 0.8796 |
| Qwen2-1.5B$_{ppl}$ | Dynamic | 0.2453 | 0.1319 | 0.0881 | 0.0643 | 0.1114 | 0.2599 | 0.8808 |
| Qwen2-7B$_{ppl}$ | Dynamic | 0.2447 | 0.1330 | 0.0891 | 0.0651 | 0.1122 | 0.2618 | 0.8817 |
| Baichuan2-7B$_{ppl}$ | Dynamic | 0.2463 | 0.1324 | 0.0887 | 0.0651 | 0.1120 | 0.2596 | 0.8811 |

As demonstrated in Table 3, PPL Chunking overlap strategy shows particularly notable performance in multi-hop QA scenarios. Specifically, except for the BERTScore metric, PPL Chunking overlap method achieves a performance gain of 2%–3% on the single-hop task. In the case of two-hop and three-hop tasks, although the rate of improvement slows slightly, a consistent gain of 0.3%–1% is maintained. Additionally, the performance across all three models exhibits an upward trend with the size of model parameters. Although the 1.5B model lags slightly behind the 7B model in terms of overall performance, it still demonstrates notable improvement over traditional chunking methods, further validating the effectiveness of PPL Chunking.

### 5.2.2 COMPARATIVE ANALYSIS OF TWO PPL CHUNKING STRATEGIES

As shown in Figure 3, we compared two PPL Chunking strategies: direct PPL Chunking and PPL Chunking with dynamic combination, both of which are effective across the CRUD dataset. Through experimental analysis, we found that the latter demonstrates superior performance. This is primarily due to direct PPL Chunking, which may result in overly long chunks, whereas the PPL Chunking with dynamic combination method effectively maintains chunk length and logical consistency.

In addition, PPL Chunking achieved significant performance improvements compared to traditional segmentation methods on BLEU series metrics and ROUGE-L. This indicates that our methods enhance the accuracy and fluency of the generated text to the reference text. Furthermore, this experiment reveals the delicate balance between model size and performance. Specifically, the performance of Qwen2-1.5B and Baichuan2-7B under this evaluation framework is closely matched, often surpassing the Qwen2-7B model across multiple metrics.

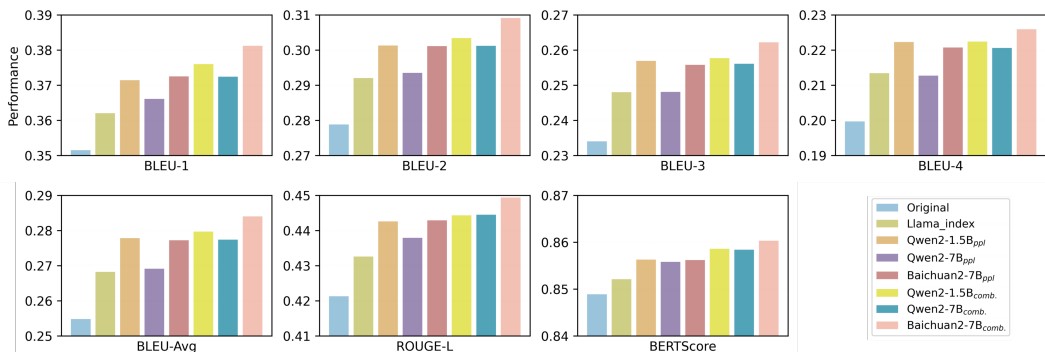

Figure 3: Performance of different methods on single-hop query in the CRUD QA dataset. *ppl* represents direct PPL Chunking, with a threshold of 0.5. *comb.* indicates PPL Chunking with dynamic combination, with a threshold of 0 when performing PPL Chunking. Precise chunk length results and performance of remaining multi-hop scenarios are included in Appendix A.3.

### 5.2.3 LONG TEXT CHUNKING AND STRATEGY SELECTION

When dealing with longer texts, we adopt the KV caching to calculate the PPL values of sentences under the premise of maintaining coherence, thereby optimizing the utilization of GPU memory and computational accuracy. Utilizing the CUAD dataset (average length 11k), we tested three models shown in Figure 4, which achieved appreciable improvements in BLEU-series metrics. Furthermore, it is noteworthy that both Qwen2-1.5B and Baichuan2-7B demonstrate comparable performance, which further confirms that the 1.5B model can maintain a impressive balance between performance and efficiency when dealing with text chunking of varying lengths.

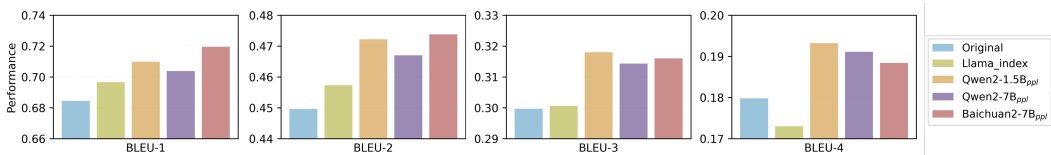

Figure 4: Performance of different methods on CUAD QA datasets. *ppl* indicates direct PPL Chunking, with a threshold of 0.

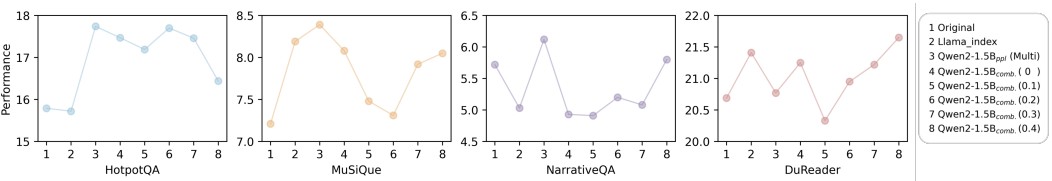

Figure 5: Performance of different methods in four long-text QA datasets of LongBench is evaluated based on F1, F1, F1, and ROUGE-L. *ppl* represents direct PPL Chunking, and *comb.* indicates PPL Chunking with dynamic combination. *Multi* represents threshold values of the parallel method in four datasets, which are 0.5, 0.5, 1.34, and 0.5 respectively, resulting in chunk lengths of 87, 90, 71, and 262 in sequence.

On the other hand, we conducted an in-depth exploration of chunking in four long-text QA datasets of LongBench, and carried out gradient experiments (0 to 0.4, step 0.1) on the threshold of PPL Chunking, aiming to reveal the intrinsic relationship between PPL distribution and chunking effectiveness. The analysis of overall PPL distribution of datasets can be found in Appendix A.4. As shown in Figure 5, when chunk length is small, the direct PPL Chunking brings greater benefits, whereas when the chunk length is longer, PPL Chunking with dynamic combination performs better. In addition, experimental results indicate that the optimal configuration of PPL Chunking relies on the PPL distribution of texts: when the PPL distribution is relatively stable, it is more appropriate to select a lower threshold (such as setting the threshold to 0 in HotpotQA, MuSiQue, and DuReader); whereas when the PPL distribution exhibits large fluctuations, choosing a higher threshold (such as setting the threshold to 0.4 in NarrativeQA) can effectively distinguish paragraphs with different information densities, improving the chunking effect. Therefore, when employing PPL for chunking, it is crucial to comprehensively consider the dual factors of chunk length and text PPL distribution to determine the relatively optimal configuration that maximizes performance.

### 5.2.4 Exploration of chunking approach for performance of Re-ranking

To explore the impact of chunking strategies on the RAG system, we evaluated the combination of different chunking and re-ranking methods. Initially, a top-10 set of relevant texts was filtered using a dense retriever. We then compared two re-ranking strategies: (1) the BgeRerank method, leveraging the bge-reranker-large model (Xiao et al., 2023), and (2) the PPLRerank method with the Qwen2-1.5B model, utilizing the re-ranking method mentioned in the coarse-grained compression section in Jiang et al. (2023).

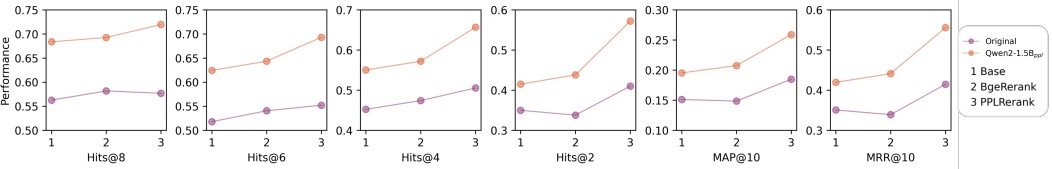

Figure 6: Performance of re-ranking strategies combined with different chunking methods in the MultiHop-RAG benchmark. *ppl* represents direct PPL Chunking, with a threshold of 0.5. The base reveals not utilizing re-ranking strategy. Precise chunk length results are included in Appendix A.5.

Experimental results (see Figure 6) revealed that PPL Chunking and PPLRerank achieved the best overall performance across all metrics. Further analysis demonstrated that, compared to traditional chunking, PPL Chunking not only provided performance gains independently but also significantly enhanced the effectiveness of the subsequent re-ranking. Notably, while traditional chunking and re-ranking strategies already deliver performance improvements, PPL Chunking resulted in even greater re-ranking gains. For instance, in the Hits@8 metric, PPLRerank under the original chunking yielded a 1.42% improvement, whereas PPLRerank under PPL Chunking achieved a 3.59% improvement.

## 6 Conclusion

This paper proposes the concept of Meta-Chunking along with its implementation strategy, namely PPL Chunking, which enable a more precise capture of the inherent logical structure of text, thereby providing a powerful tool for optimizing text segmentation within the RAG pipeline. To balance the effectiveness of fine-grained and coarse-grained text segmentation, we present a dynamic combination approach with Meta-Chunking to address the limitation when dealing with diverse texts. Our comprehensive evaluation using multiple metrics on eleven datasets demonstrates that Meta-Chunking significantly outperforms both rule-based and similarity-based chunking, while also achieving a better balance between performance, time cost, and computational cost compared to current LLMs approaches.

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

## A APPENDIX

### A.1 THEORETICAL PROOF FOR PPL CHUNKING

Firstly, we illustrate the relationship between cross-entropy and two distributions $P$ and $Q$ in another way. Based on sequencing inequality

$$\sum_{i=1}^{n} a_i b_i \geq \sum_{i=1}^{n} a_i b_{j(i)} \geq \sum_{i=1}^{n} a_i b_{n+1-i}$$

where $a_1 \geq a_2 \geq \cdots \geq a_n$, $b_1 \geq b_2 \geq \cdots \geq b_n$ and $(j(1), j(2), \ldots, j(n))$ is an arbitrary sorting of $(1, 2, \ldots, n)$, it can be observed that the sum of products of larger numbers paired together is the maximum, while the sum of products of larger numbers paired with smaller numbers is the minimum. We desire the cross-entropy $H(P, Q)$ to be as small as possible, which means that when $P(x)$ is relatively large, $-\log Q(x)$ should be relatively small, thereby resulting in $Q(x)$ also being relatively large. Therefore, a smaller cross-entropy indicates that the prediction is closer to the actual label.

Afterwards, inspired by insights provided in Huyen (2019), a property of formula (7) is proved: $G_{K+1} \leq G_K$ for all $K \geq 1$.

*Proof.*

$$G_K - G_{K+1}$$
$$= -\sum_{T_k} P(T_k) \log_a P(t_k|T_{k-1}) + \sum_{T_{k+1}} P(T_{k+1}) \log_a P(t_{k+1}|T_k)$$
$$= \sum_{T_{k-1}} \left[ \sum_{t_k, t_{k+1}} P(T_{k+1}) \log_a P(t_{k+1}|T_k) - \sum_{t_k} P(T_k) \log_a P(t_k|T_{k-1}) \right]$$
$$\geq \sum_{T_{k-1}} \left[ \sum_{t_k, t_{k+1}} P(T_{k+1}) \log_a P(t_{k+1}|T_{k-1}) - \sum_{t_k} P(T_k) \log_a P(t_k|T_{k-1}) \right]$$
$$= \sum_{T_{k-1}} \left[ \sum_{t_k, t_{k+1}} P(T_{k-1}, t_k, t_{k+1}) \log_a P(t_{k+1}|T_{k-1}) - \sum_{t_k} P(T_{k-1}, t_k) \log_a P(t_k|T_{k-1}) \right]$$
$$= \sum_{T_{k-1}} \left[ \sum_{t_{k+1}} \log_a P(t_{k+1}|T_{k-1}) \sum_{t_k} P(T_{k-1}, t_k, t_{k+1}) - \sum_{t_k} P(T_{k-1}, t_k) \log_a P(t_k|T_{k-1}) \right]$$
$$= \sum_{T_{k-1}} \left[ \sum_{t_{k+1}} P(T_{k-1}, t_{k+1}) \log_a P(t_{k+1}|T_{k-1}) - \sum_{t_k} P(T_{k-1}, t_k) \log_a P(t_k|T_{k-1}) \right]$$
$$= 0$$

The reason for the last equality is that $t_{k+1}$ and $t_k$ belong to the same domain. Thus, the proof is complete. $\square$

Eventually, we illustrate bounds of entropy, so as to demonstrate the positive correlation between $H(P, Q)$ and $D_{KL}(P||Q)$ in formula (3).

*Proof.* Let $P$ be a discrete random variable with a finite range of values denoted by $W := \{w_1, w_2, \ldots, w_l\}$. Set $p_i = P\{P = w_i\}$ for $i = 1, 2, \ldots, l$, and assume that $p_i > 0$ for all $i \in \{1, 2, \ldots, l\}$. According to Lemma 2 in Dragomir & Goh (1997), if

$$\gamma := \max_{i,j} \frac{\theta_i}{\theta_j} \leq \varphi(\varepsilon) := 1 + \varepsilon \ln c + \sqrt{\varepsilon \ln c(\varepsilon \ln c + 2)}$$

then

$$0 \leq \log_c \left( \sum_{k=1}^{l} p_k \theta_k \right) - \sum_{k=1}^{l} p_k \log_c \theta_k \leq \varepsilon$$

where $\theta_k \in (0, +\infty)$, $p_k \geq 0$ with $\sum_{k=1}^{l} p_k = 1$ and $c > 1$. Given that $\theta_k = 1/p_k$, the aforementioned inequality can be transformed into

$$0 \leq \log_c l - H_c(P) \leq \varepsilon$$

where $\varepsilon > 0$ satisfies the following conditions

$$\max_{i,j} \frac{p_i}{p_j} \leq \varphi(\varepsilon)$$

Furthermore, we can derive bounds for entropy as $\log_c l - \varepsilon \leq H_c(P) \leq \log_c l$. The proof is concluded. $\square$

## A.2 MAIN EXPERIMENTAL DETAILS

All language models utilized in this paper employ the chat or instruct versions where multiple versions exist, and are loaded in full precision (Float32). The vector database is constructed using Milvus, where the embedding model for English texts is bge-large-en-v1.5, and bge-base-zh-v1.5 for Chinese texts. When conducting QA, the system necessitates dense retrievals from the vector database, with top_k set to 8 for CRUD and RAGBench, 10 for MultiHop-RAG, and 5 for Long-Bench.

In experiments, we utilized a total of four benchmarks, and their specific configurations are detailed as follows:

(a) **Rule-based Chunking Methods**

- **Original**: This method divides long texts into segments of a fixed length, such as two hundred Chinese characters or words, without considering sentence boundaries.
- **Llama_index** (Langchain, 2023): This method considers both sentence completeness and token counts during segmentation. It prioritizes maintaining sentence boundaries while ensuring that the number of tokens in each chunk are close to a preset threshold. We use the `SimpleNodeParser` function from `Llama_index`, adjusting the `chunk_size` parameter to control segment length. Overlaps are handled by dynamically overlapping segments using the `chunk_overlap` parameter, ensuring sentence completeness during segmentation and overlapping.

(b) **Dynamic Chunking Methods**

- **Similarity Chunking** (Xiao et al., 2023): Utilizes pre-trained sentence embedding models to calculate the cosine similarity between sentences. By setting a similarity threshold, sentences with lower similarity are selected as segmentation points, ensuring that sentences within each chunk are highly semantically related. This method employs the `SemanticSplitterNodeParser` from `Llama_index`. For English texts, we exploit the bge-large-en-v1.5 model, and for Chinese texts, the bge-base-zh-v1.5 model. The size of the text chunks is controlled by adjusting the similarity threshold.
- **LumberChunker** (Duarte et al., 2024): Leverages the reasoning capabilities of LLMs to predict suitable segmentation points within the text. We utilize Qwen2 models with 1.5B and 7B parameters, set to full precision.
- **Dense X Retrieval** (Chen et al., 2023a): Introduces a new retrieval granularity called propositions, which condenses and segments text by training an information extraction model.

In order to control variables during the experiment, we ensured that each dataset had approximately the same size when divided into chunks using different methods. The specific chunk lengths and corresponding thresholds for each dataset in the main experiment are shown in Table 4. We first explored direct chunking of Qwen2-72B, using the prompt displayed in Table 5, and found that it took too long. We then exploited this as a comparison to explore other methods.

Table 4: Chunk length and corresponding threshold settings for different methods. - indicates no relevant setting is involved. The first four datasets are sourced from LongBench. *0+comb.* signifies that an initial chunking is performed using a threshold of 0, followed by a dynamic combination approach to derive the final chunks. In Llama_index and Qwen2-72B, *a(b)* indicates that the chunk size of *a* can be achieved by setting the chunking parameter to *b*. For other instances of *a(b)*, it represents the dynamic combination of chunks where setting the combination length to *b* results in a final chunk size of *a*.

| Dataset | 2WikiMultihopQA | | Qasper | | MultiFieldQA-en | | MultiFieldQA-zh | | MultiHop-RAG | |
| Chunking Method | Length | Threshold | Length | Threshold | Length | Threshold | Length | Threshold | Length | Threshold |
|---|---|---|---|---|---|---|---|---|---|---|
| *Baselines with rule-based or similarity-based chunking* | | | | | | | | | | |
| Original | 123 | - | 121 | - | 113 | - | 178 | - | 78 | - |
| Llama_index | 122.61(215) | - | 120.91(198) | - | 112.59(208) | - | 178.04(242) | - | 79.68 | - |
| Similarity Chunking | 125.24 | 0.82 | 122.91 | 0.83 | 114.18 | 0.83 | 180.23 | 0.73 | 80.13 | 0.75 |
| *LLMs Direct Chunking* | | | | | | | | | | |
| Qwen2-72B | 122.13(128) | - | 120.17(90) | - | 111.98(88) | - | 178.05(190) | - | - | - |
| *Chunking based on Qwen2-0.5B* | | | | | | | | | | |
| Qwen2-0.5B$_{sent.}$ | 122.33(148) | 0+comb. | 120.07(147) | 0+comb. | 112.46(136) | 0+comb. | 178.09(180) | 0+comb. | 78.04(91) | 0+comb. |
| Qwen2-0.5B$_{comb.}$ | 122.39(152) | 0+comb. | 120.04(155) | 0+comb. | 112.30(139) | 0+comb. | 178.36(160) | 0+comb. | 78.17(89) | 0+comb. |
| *Chunking based on Qwen2-1.5B* | | | | | | | | | | |
| Qwen2-1.5B$_{chunk}$ | 121.99(148) | 0+comb. | 120.21(144) | 0+comb. | 111.52(134) | 0+comb. | 177.80(200) | 0+comb. | 78.16(97) | 0+comb. |
| Qwen2-1.5B$_{comb.}$ | 122.48(152) | 0+comb. | 120.56(156) | 0+comb. | 111.35(138) | 0+comb. | 178.00(159) | 0+comb. | 78.19(89) | 0+comb. |
| *Chunking based on Qwen2-7B* | | | | | | | | | | |
| Qwen2-7B$_{chunk}$ | 121.81(138) | 0+comb. | 120.01(141) | 0+comb. | 111.56(129) | 0+comb. | 178.00(188) | 0+comb. | 77.49(95) | 0+comb. |
| Qwen2-7B$_{comb.}$ | 122.26(152) | 0+comb. | 120.26(155) | 0+comb. | 111.47(137) | 0+comb. | 177.80(156) | 0+comb. | 78.11(89) | 0+comb. |
| Qwen2-7B-base$_{comb.}$ | 122.34(152) | 0+comb. | 120.43(155) | 0+comb. | 112.76(139) | 0+comb. | - | - | - | - |

Table 5: Prompt for direct chunking of Qwen2-72B.

---

**Chunking Prompt**

You are an expert in text segmentation, tasked with dividing given text into blocks. You must adhere to the following four conditions:

1. Aim to keep each block around 128 English words in length.

2. Segment the text based solely on its logical and semantic structures.

3. Do not alter the original vocabulary or structure of the text.

4. Do not add any new words or symbols.

By solely determining the boundaries for text segmentation, divide the original text into blocks and output them individually, separated by a clear delimiter '− − −Block Separator − − −'. Do not output any other explanations. If you understand, please proceed to segment the following text into blocks: [Text to be segmented]

---

In the Margin Sampling Chunking method, we also use prompt, which mainly consists of two parts: instructions for guiding LLMs to perform chunking and two segmentation schemes. The specific form is shown in Table 6.

Table 6: Prompt used in Margin Sampling Chunking.

---

**Chunking Prompt**

This is a text chunking task. You are a text analysis expert. Please choose one of the following two options based on the logical structure and semantic content of the provided sentence:

1. Split *sentence1+sentence2* into *sentence1* and *sentence2* two parts;

2. Keep *sentence1+sentence2* unsplit in its original form;

Please answer 1 or 2.

---

### A.3 CHUNKING SITUATIONS OF THE CRUD DATASET

#### A.3.1 FILTERING OF CORPORA RELATED TO QA TASKS IN THE CRUD DATASET

In this experiment, we selected three QA datasets from the CRUD benchmark. Among them, the single-hop QA dataset consists of questions focused on extracting factual information from a single document. These questions typically require precise retrieval of specific details such as dates, individuals, or events from the provided text. The two-hop QA dataset, on the other hand, evaluates integration capabilities and understanding of informational relationships between different documents. The more complex three-hop QA dataset often presents more intricate questions, demanding LLMs to process a greater number of information sources to formulate a complete and accurate response.

Before the chunking phase, we collected original news articles used in all types of QA tasks in CRUD. Specifically, since CRUD provides evidence context snippets relied on by each QA pair, as well as the original news library where the context snippets are extracted, we can obtain the original news articles containing the context snippets through sentence matching. Taking the two-hop QA as an example, CRUD provides two news snippets, *news1* and *news2*, which are necessary to answer *questions*. We then save the matched original news articles *matched_news1* and *matched_news2* that contain *news1* and *news2*. Finally, from the original news library of 80,000 articles, we recall all 10,000 news articles containing context snippets as the initial text for chunking.

#### A.3.2 EXPERIMENTAL SETUP FOR TWO RESEARCHES BASED ON THE CRUD DATASET

We conducted two sets of experiments with overlapping and non-overlapping chunking on the CRUD dataset, respectively in Section 5.2.1 and 5.2.2. The chunk length and overlap length are shown in Table 7. Additionally, the specific values for the bar chart presented in Figure 3 are detailed in Table 8.

Table 7: Settings of overlap length and chunk length for different chunking methods in the CRUD dataset. *ppl* represents direct PPL Chunking, with a threshold of 0.5. *comb.* indicates PPL Chunking with dynamic combination, with a threshold of 0 when performing PPL Chunking.

| Chunking Method | Overlap Length | Chunk Length |
|---|---|---|
| *Chunking with Overlap* | | |
| Original | 50 | 218 |
| Llama_index | 48.78 | 217.03 |
| Qwen2-1.5B$_{ppl}$ | 49.97 | 212.79 |
| Qwen2-7B$_{ppl}$ | 50.41 | 217.53 |
| Baichuan2-7B$_{ppl}$ | 48.91 | 201.35 |
| *Chunking without Overlap* | | |
| Original | 0 | 179 |
| Llama_index | 0 | 177.53 |
| Qwen2-1.5B$_{ppl}$ | 0 | 173.88 |
| Qwen2-7B$_{ppl}$ | 0 | 178.59 |
| Baichuan2-7B$_{ppl}$ | 0 | 162.56 |
| Qwen2-1.5B$_{comb.}$ | 0 | 177.95 |
| Qwen2-7B$_{comb.}$ | 0 | 178.09 |
| Baichuan2-7B$_{comb.}$ | 0 | 178.09 |

Further analysis demonstrates that in single-hop and double-hop query scenarios presented in Table 8, PPL Chunking achieved significant performance improvements compared to traditional chunking methods on BLEU series metrics and ROUGE-L. This indicates that our methods enhance the accuracy and fluency of the generated text to the reference text. However, the relatively smaller margin

Table 8: Performance of different methods on the CRUD QA dataset. *ppl* represents direct PPL Chunking, with a threshold of 0.5. *comb.* indicates PPL Chunking with dynamic combination, with a threshold of 0 when performing PPL Chunking.

| Chunking Method | BLEU-1 | BLEU-2 | BLEU-3 | BLEU-4 | BLEU-Avg | ROUGE-L | BERTScore |
|---|---|---|---|---|---|---|---|
| *Single-hop Query* | | | | | | | |
| Original | 0.3515 | 0.2788 | 0.2340 | 0.1997 | 0.2548 | 0.4213 | 0.8489 |
| Llama_index | 0.3620 | 0.2920 | 0.2480 | 0.2134 | 0.2682 | 0.4326 | 0.8521 |
| Qwen2-1.5B$_{ppl}$ | 0.3714 | 0.3013 | 0.2569 | 0.2223 | 0.2778 | 0.4426 | 0.8563 |
| Qwen2-7B$_{ppl}$ | 0.3661 | 0.2935 | 0.2481 | 0.2127 | 0.2691 | 0.4379 | 0.8558 |
| Baichuan2-7B$_{ppl}$ | 0.3725 | 0.3011 | 0.2558 | 0.2207 | 0.2772 | 0.4429 | 0.8562 |
| Qwen2-1.5B$_{comb.}$ | 0.3760 | 0.3034 | 0.2577 | 0.2224 | 0.2797 | 0.4443 | 0.8586 |
| Qwen2-7B$_{comb.}$ | 0.3724 | 0.3012 | 0.2561 | 0.2206 | 0.2774 | 0.4445 | 0.8584 |
| Baichuan2-7B$_{comb.}$ | 0.3812 | 0.3091 | 0.2622 | 0.2259 | 0.2840 | 0.4494 | 0.8603 |
| *Two-hop Query* | | | | | | | |
| Original | 0.2322 | 0.1324 | 0.0919 | 0.0695 | 0.1133 | 0.2613 | 0.8768 |
| Llama_index | 0.2315 | 0.1321 | 0.0923 | 0.0697 | 0.1133 | 0.2585 | 0.8762 |
| Qwen2-1.5B$_{ppl}$ | 0.2328 | 0.1326 | 0.0918 | 0.0694 | 0.1133 | 0.2611 | 0.8749 |
| Qwen2-7B$_{ppl}$ | 0.2310 | 0.1323 | 0.0916 | 0.0691 | 0.1124 | 0.2597 | 0.8752 |
| Baichuan2-7B$_{ppl}$ | 0.2350 | 0.1341 | 0.0924 | 0.0695 | 0.1141 | 0.2637 | 0.8772 |
| Qwen2-1.5B$_{comb.}$ | 0.2372 | 0.1363 | 0.0950 | 0.0722 | 0.1164 | 0.2658 | 0.8743 |
| Qwen2-7B$_{comb.}$ | 0.2364 | 0.1360 | 0.0945 | 0.0713 | 0.1161 | 0.2661 | 0.8761 |
| Baichuan2-7B$_{comb.}$ | 0.2325 | 0.1329 | 0.0917 | 0.0689 | 0.1133 | 0.2623 | 0.8754 |
| *Three-hop Query* | | | | | | | |
| Original | 0.2494 | 0.1317 | 0.0869 | 0.0636 | 0.1110 | 0.2595 | 0.8827 |
| Llama_index | 0.2464 | 0.1327 | 0.0883 | 0.0644 | 0.1120 | 0.2596 | 0.8840 |
| Qwen2-1.5B$_{ppl}$ | 0.2402 | 0.1260 | 0.0827 | 0.0596 | 0.1054 | 0.2531 | 0.8802 |
| Qwen2-7B$_{ppl}$ | 0.2415 | 0.1266 | 0.0828 | 0.0597 | 0.1058 | 0.2549 | 0.8816 |
| Baichuan2-7B$_{ppl}$ | 0.2460 | 0.1293 | 0.0851 | 0.0615 | 0.1084 | 0.2568 | 0.8828 |
| Qwen2-1.5B$_{comb.}$ | 0.2449 | 0.1294 | 0.0855 | 0.0624 | 0.1086 | 0.2566 | 0.8828 |
| Qwen2-7B$_{comb.}$ | 0.2408 | 0.1274 | 0.0837 | 0.0610 | 0.1068 | 0.2551 | 0.8825 |
| Baichuan2-7B$_{comb.}$ | 0.2494 | 0.1324 | 0.0870 | 0.0632 | 0.1111 | 0.2613 | 0.8832 |

of improvement observed on the BERTScore, a BERT-based semantic similarity evaluation metric, may reflect a lower sensitivity of deep semantic understanding to chunking, as well as the limitations of the current BERTScore models in capturing precise semantics.

Finally, for three-hop query, although the performance of Qwen2-1.5B and Qwen2-7B using PPL Chunking was slightly lower than traditional methods, Baichuan2-7B performed comparably. However, when chunk overlap is introduced, the PPL Chunking method exhibits positive changes (as shown in Tables 3). This suggests that the effectiveness of segmentation strategies may be jointly influenced by query complexity and text characteristics.

## A.4  CHUNKING SITUATIONS OF LONG TEXT DATASETS

We also conducted experiments on longer datasets. According to corresponding expressions in benchmarks, the average length of the CUAD dataset is 11k, and average lengths of four datasets in MultiHop-RAG are 9k, 11k, 18k, and 16k. The chunk lengths of these two sets of experiments are shown in Tables 9 and 10. Additionally, the specific values presented in Figures 4 and 5 correspond to Tables 11 and 12.

According to Table 10, it can be observed that HotpotQA, MuSiQue, and DuReader achieve a suitable chunk length with a lower threshold, while NarrativeQA only reaches it when the threshold

Table 9: Settings of overlap length and chunk length for different chunking methods in the CUAD dataset. *ppl* represents direct PPL Chunking, with a threshold of 0.

| Chunking Method | Overlap Length | Chunk Length |
|---|---|---|
| Original | 0 | 98.00 |
| Llama_index | 0 | 98.49 |
| Qwen2-1.5B$_{ppl}$ | 0 | 97.70 |
| Qwen2-7B$_{ppl}$ | 0 | 96.08 |
| Baichuan2-7B$_{ppl}$ | 0 | 97.59 |

Table 10: Chunk length and corresponding threshold settings for different chunking methods in four long-text QA datasets of LongBench. - indicates no relevant setting. In Llama_index, *a(b)* represents that a chunk length of *a* can be obtained by setting the chunking parameter to *b*. The remaining *a(b)* indicates that a final chunk length of *a* is obtained by setting the combination length to *b*.

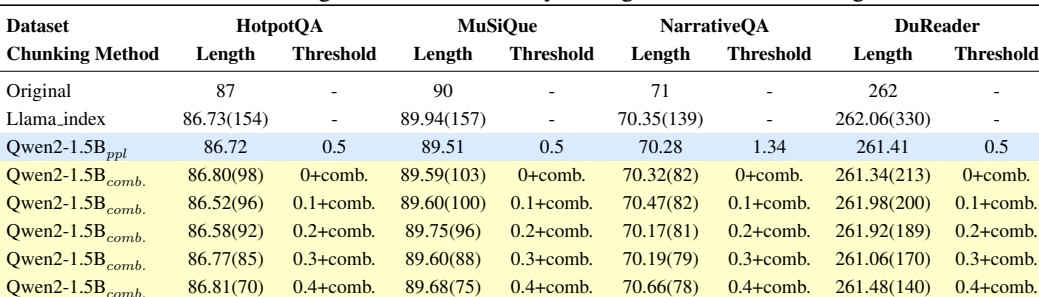

| Dataset | HotpotQA | | MuSiQue | | NarrativeQA | | DuReader | |
|---|---|---|---|---|---|---|---|---|
| Chunking Method | Length | Threshold | Length | Threshold | Length | Threshold | Length | Threshold |
| Original | 87 | - | 90 | - | 71 | - | 262 | - |
| Llama_index | 86.73(154) | - | 89.94(157) | - | 70.35(139) | - | 262.06(330) | - |
| Qwen2-1.5B$_{ppl}$ | 86.72 | 0.5 | 89.51 | 0.5 | 70.28 | 1.34 | 261.41 | 0.5 |
| Qwen2-1.5B$_{comb.}$ | 86.80(98) | 0+comb. | 89.59(103) | 0+comb. | 70.32(82) | 0+comb. | 261.34(213) | 0+comb. |
| Qwen2-1.5B$_{comb.}$ | 86.52(96) | 0.1+comb. | 89.60(100) | 0.1+comb. | 70.47(82) | 0.1+comb. | 261.98(200) | 0.1+comb. |
| Qwen2-1.5B$_{comb.}$ | 86.58(92) | 0.2+comb. | 89.75(96) | 0.2+comb. | 70.17(81) | 0.2+comb. | 261.92(189) | 0.2+comb. |
| Qwen2-1.5B$_{comb.}$ | 86.77(85) | 0.3+comb. | 89.60(88) | 0.3+comb. | 70.19(79) | 0.3+comb. | 261.06(170) | 0.3+comb. |
| Qwen2-1.5B$_{comb.}$ | 86.81(70) | 0.4+comb. | 89.68(75) | 0.4+comb. | 70.66(78) | 0.4+comb. | 261.48(140) | 0.4+comb. |

is set to 1.34. This indicates that PPL distribution of the first three datasets is relatively flat with small oscillations, whereas NarrativeQA exhibits significant fluctuations. Considering the chunking performance presented in Table 12, it suggests that direct PPL Chunking is more suitable when chunk length is small, while the combination of PPL Chunking and dynamic merging is preferable for larger chunk lengths. Furthermore, regarding the approach of PPL Chunking with dynamic combination, it is more appropriate to select a smaller threshold when the PPL amplitude is small, and a larger threshold when the PPL amplitude is significant.

Table 11: Performance of different methods on CUAD QA datasets. *ppl* indicates direct PPL Chunking, with a threshold of 0.

| Chunking Method | BLEU-1 | BLEU-2 | BLEU-3 | BLEU-4 | BLEU-Avg | ROUGE-L | BERTScore |
|---|---|---|---|---|---|---|---|
| Original | 0.6845 | 0.4496 | 0.2997 | 0.1798 | 0.3513 | 0.4217 | 0.8043 |
| Llama_index | 0.6966 | 0.4573 | 0.3006 | 0.1730 | 0.3493 | 0.4137 | 0.8001 |
| Qwen2-1.5B$_{ppl}$ | 0.7098 | 0.4722 | 0.3180 | 0.1932 | 0.3677 | 0.4060 | 0.8006 |
| Qwen2-7B$_{ppl}$ | 0.7038 | 0.4670 | 0.3143 | 0.1911 | 0.3638 | 0.4070 | 0.8018 |
| Baichuan2-7B$_{ppl}$ | 0.7195 | 0.4738 | 0.3160 | 0.1884 | 0.3665 | 0.4111 | 0.8025 |

A.5   EXPERIMENTAL SETUPS FOR EXPLORING THE IMPACT OF CHUNKING ON RE-RANKING

Tables 13 and 14 present chunk lengths that need to be set for Figure 6 and the specific values for drawing, respectively. Focusing on this batch of experiments, we first retrieve 10 relevant text chunks for each question through a dense retriever, and then applied various re-ranking methods for secondary sorting to analyze changes in recall performance.

Table 12: Performance of different methods in four long-text QA datasets of LongBench. *ppl* represents direct PPL Chunking, and *comb.* indicates PPL Chunking with dynamic combination. *Multi* represents threshold values of the parallel method in four datasets, which are 0.5, 0.5, 1.34, and 0.5 respectively, resulting in chunk lengths of 87, 90, 71, and 262 in sequence.

| Chunking Method | Dataset Threshold | HotpotQA F1 | MuSiQue F1 | NarrativeQA F1 | DuReader ROUGE-L |
|---|---|---|---|---|---|
| Original | - | 15.79 | 7.21 | 5.72 | 20.69 |
| Llama_index | - | 15.72 | 8.19 | 5.03 | 21.41 |
| Qwen2-1.5B$_{ppl}$ | *Multi* | **17.74** | **8.39** | **6.12** | 20.77 |
| Qwen2-1.5B$_{comb.}$ | 0 | 17.47 | 8.08 | 4.93 | 20.77 |
| Qwen2-1.5B$_{comb.}$ | 0.1 | 17.19 | 7.48 | 4.91 | 20.33 |
| Qwen2-1.5B$_{comb.}$ | 0.2 | 17.70 | 7.31 | 5.20 | 20.95 |
| Qwen2-1.5B$_{comb.}$ | 0.3 | 17.46 | 7.92 | 5.08 | 21.22 |
| Qwen2-1.5B$_{comb.}$ | 0.4 | 16.44 | 8.05 | 5.80 | **21.65** |

Table 13: Chunk length and its corresponding threshold settings when exploring the impact of chunking on re-ranking. - indicates no relevant setting.

| Chunking and Re-ranking | Chunk Length | Threshold |
|---|---|---|
| Original | 78 | - |
| Original and BgeRerank | 78 | - |
| Original and PPLRerank | 78 | - |
| Qwen2-1.5B$_{ppl}$ | 77.60 | 0.5 |
| Qwen2-1.5B$_{ppl}$ and BgeRerank | 77.60 | 0.5 |
| Qwen2-1.5B$_{ppl}$ and PPLRerank | 77.60 | 0.5 |

Table 14: Performance of re-ranking strategies combined with different chunking methods in the MultiHop-RAG benchmark. *ppl* represents direct PPL Chunking, with a threshold of 0.5.

| Chunking and Re-ranking | Hits@8 | Hits@6 | Hits@4 | Hits@2 | MAP@10 | MRR@10 |
|---|---|---|---|---|---|---|
| Original | 0.5627 | 0.5180 | 0.4523 | 0.3499 | 0.1512 | 0.3507 |
| Original and BgeRerank | **0.5818** | 0.5406 | 0.4741 | 0.3379 | 0.1486 | 0.3391 |
| Original and PPLRerank | 0.5769 | **0.5521** | **0.5055** | **0.4102** | **0.1849** | **0.4147** |
| Qwen2-1.5B$_{ppl}$ | 0.6838 | 0.6244 | 0.5503 | 0.4151 | 0.1954 | 0.4195 |
| Qwen2-1.5B$_{ppl}$ and BgeRerank | 0.6927 | 0.6435 | 0.5721 | 0.4381 | 0.2075 | 0.4413 |
| Qwen2-1.5B$_{ppl}$ and PPLRerank | **0.7197** | **0.6931** | **0.6568** | **0.5721** | **0.2590** | **0.5558** |

