# OpenReview forum: "Meta-Chunking: Learning Efficient Text Segmentation via Logical Perception"
_ICLR.cc/2025/Conference — ICLR 2025 Conference Withdrawn Submission_

### Official Review · Reviewer_y1mZ · 2024-10-25

**Soundness:** 2
**Presentation:** 1
**Contribution:** 2
**Rating:** 3
**Confidence:** 4

**Summary:**

The authors propose some new text chunking methods in this paper: Margin Sampling Chunking and Perplexity (PPL) Chunking. Margin sampling chunking prompts LLM to answer whether two sentences should be in the same chunk and PPL chunking uses the PPL distribution in the text series to find chunking points. The paper shows the advantage of the proposed methods over some simple baselines on retrieval efficiency and accuracy.

**Strengths:**

The proposed PPL chunking is intuitive and utilizes the reflection by text relations by PPL.

The experiments are somehow comprehensive.

**Weaknesses:**

The contributions are overclaimed. From my viewpoint, marginal sampling chunking is similar to a simplification of the mentioned LumberChunker, which both prompts LLM to find chunking points. The only improvement is the design of a simpler prompt to reduce the language capability requirement, which is quite incremental.

For the PPL chunking, the authors do not clarify their specific design to locate the chucking points, which are quite complex in (3), but they do not have enough explanation or empirical experiment support.

The experiment presentation is confusing, the authors enumerate a lot of language models but many of them are missed in the PPL chunker rows. It should be recognized with different chunking methods applied to the same language model. Also, the performance of prompt-based chunkers is highly influenced by the prompt design while the PPL chunker is not. Given the rather similar performance between the two paradigms, the advantage of PPL chunker is questionable considering the prompt selection

The writing is also problematic, the introduction does not clearly emphasize the reason why text chunking is important. As a relatively new field, the related work section is placed at the end of the paper, hindering the reader's understanding of this emerging field. Formulas are also not well defined like (1) and (3).

**Questions:**

The perplexity is highly biased to the words in the sentence. For instance, stop words like "the", punctuations, and subwords like "-soft" after "Micro" will have very low perplexity, how will you handle such bias?

Calculating PPL in your experiments does not require strong instruction-following ability, do you think language models before SFT can perform well as a PPL chunker?

The RAG performance is a synergy between LLM and the retriever, for instance, LLM with stronger long-context ability will favor larger chunks, how will your proposed method remain useful with the development of LLM ability?

---

> ### Author Response · Authors · 2024-11-24
>
> Thank you for your valuable feedback, which helps us improve the presentation of our paper.
>
>  > W1: The contributions are overclaimed. From my viewpoint, marginal sampling chunking is similar to a simplification of the mentioned LumberChunker, which both prompts LLM to find chunking points. The only improvement is the design of a simpler prompt to reduce the language capability requirement, which is quite incremental.
>
> - Our primary innovation lies in the introduction of the PPL Chunking, addressing the issues of prolonged runtime and high instruction-following capability requirements inherent in current LLMs chunking methodologies. Our research focuses on whether models of 7B or even smaller sizes can effectively and rapidly perform chunking. Given that LumberChunker is virtually inapplicable to models of 7B parameters or less, it becomes challenging to compare different methods within the same model scale. Consequently, we first propose Margin Sampling Chunking as a solution to alleviate the high instruction-following demands of the sota method, serving as an intermediary step to facilitate comparison with PPL Chunking.
>
> - The structure of our previous paper led to some misunderstandings, prompting us to reorganize the methodology and main experimental sections. Additionally, we have relocated the content pertaining to Margin Sampling Chunking from the methodology section to the baseline section, aligning with the changes in the primary experiments. It is important to note that Margin Sampling Chunking is not merely an enhancement of the Prompt. To adapt the method for smaller models, we devised a margin sampling strategy to determine whether chunking should be performed. This differs from the LumberChunker approach, which relies on regular expression matching within answers generated by LLMs, thereby necessitating a high level of model instruction-following capability. In experiments, models of 7B parameters or less struggle with such complex tasks, resulting in chaotic outputs that ultimately impede the chunking process.
>
>  > W2: For the PPL chunking, the authors do not clarify their specific design to locate the chucking points, which are quite complex in (3), but they do not have enough explanation or empirical experiment support.
>
> This formula serves as a guide for the PPL Chunking to find minimum points of the PPL distribution: when the PPL on both sides of a point are higher than at that point, and the difference on at least one side exceeds the preset threshold $\theta$; or when the difference between the left point and the point is greater than $\theta$ and the right point equals the point value. These minima are regarded as potential chunk boundaries. We provide supplementary explanations for this in lines 182 to 185 of the paper. Since this formula is used to guide the PPL chunking method in identifying segmentation points, it is applied in all experiments related to PPL chunking, demonstrating the feasibility of this approach.
>
>  > W3: The experiment presentation is confusing, the authors enumerate a lot of language models but many of them are missed in the PPL chunker rows. It should be recognized with different chunking methods applied to the same language model. Also, the performance of prompt-based chunkers is highly influenced by the prompt design while the PPL chunker is not. Given the rather similar performance between the two paradigms, the advantage of PPL chunker is questionable considering the prompt selection
>
> - As stated in W1, we restructured the main results section to provide a clearer presentation of the PPL Chunking method, which addresses the limitations of current sota method that is either inapplicable to small models or suffer from long runtime. By employing three models of different scales, namely Qwen2-0.5B, Qwen2-1.5B, and Qwen2-7B, we investigated the performance and time consumption of various methods on these models.
>
> - Compared to similarity-based chunking and LLMs chunking methods, PPL chunking maintains comparable performance while achieving faster speeds, making its runtime more acceptable in practical scenarios. Margin Sampling Chunking primarily addresses the high instruction-following capability requirements of the LumberChunker method. However, both of these LLMs chunking methods consume significant resources and time, posing challenges for practical applications. Although PPL Chunking is also a LLM method, its design enhances chunking speed, making it better suited for real-world scenarios involving massive text data.

---

> > ### Author Response · Authors · 2024-11-24
> >
> > > W4: The writing is also problematic, the introduction does not clearly emphasize the reason why text chunking is important. As a relatively new field, the related work section is placed at the end of the paper, hindering the reader's understanding of this emerging field. Formulas are also not well defined like (1) and (3).
> >
> > 1. In the introduction, while discussing the necessity of RAG and the limitations of existing chunking methods within the pipeline, we neglected to explain why chunking is required. To rectify this, we have included supplementary explanations in lines 47-52 of the introduction section.
> >
> >     - By delicately splitting long documents into multiple chunks, this module not only significantly improves the processing efficiency and performance of the system, reducing the consumption of computing resources, but also enhances the accuracy of retrieval [1]. Meanwhile, the chunking strategy allows information to be more concentrated, minimizing the interference of irrelevant information, enabling LLMs to focus more on the specific content of each text chunk and generate more precise responses [2].
> >
> > 2. To present the relevant knowledge more clearly, we have moved the related work section to the front. Additionally, we provided further explanations for some of the formulas presented in the paper, aiming to facilitate a deeper understanding of the methodologies and calculations involved.
> >
> >
> > [1] Maciej Besta, et al. Multi-head rag: Solving multi-aspect problems with llms. arXiv preprint arXiv:2406.05085, 2024.
> >
> > [2] Weihang Su, et al. Dragin: Dynamic retrieval augmented generation based on the real-time information needs of large language models. arXiv
> > preprint arXiv:2403.10081, 2024.

---

> > > ### Author Response · Authors · 2024-11-24
> > >
> > > Thank you for your thoughtful questions. We hope to resolve your concerns by providing clarifications regarding our experiments and methodologies.
> > >
> > > > Q1: The perplexity is highly biased to the words in the sentence. For instance, stop words like "the", punctuations, and subwords like "-soft" after "Micro" will have very low perplexity, how will you handle such bias?
> > >
> > > In real-world scenarios, a sentence should generally not be segmented, as this would lead to a disruption in semantics. When calculating perplexity values, we also consider the sentence as a whole, as shown in Formula 1, where we compute the average perplexity of all tokens within a sentence to mitigate the impact of special cases on the calculation.
> > >
> > > > Q2: Calculating PPL in your experiments does not require strong instruction-following ability, do you think language models before SFT can perform well as a PPL chunker?
> > >
> > > To address this issue, we conducted an additional set of experiments to compare the performance of PPL Chunking using both the base model and the instruct model, as shown in Table 2. Our findings indicate that PPL Chunking does not impose strict requirements on the model's ability to follow specific instructions, with the base model and the instruct model exhibiting comparable performance. This further demonstrates the adaptability of PPL Chunking.
> > >
> > > > Q3: The RAG performance is a synergy between LLM and the retriever, for instance, LLM with stronger long-context ability will favor larger chunks, how will your proposed method remain useful with the development of LLM ability?
> > >
> > > - To address the issue of diverse chunking in real-world scenarios, we proposed a strategy that combines meta-chunking with dynamic combination in line 99 of the paper, aiming to achieve a balance between fine-grained and coarse-grained text chunking. Dynamic combination primarily plays an auxiliary role in the chunking process, dynamically merging meta-chunks according to users' chunk length requirements to facilitate subsequent retrieval and generation.
> > >
> > > - This paper primarily proposes the PPL Chunking method to realize meta-chunking, ensuring that each segmented chunk contains a complete and independent logical expression. It leverages the hallucinations of language models to perceive text boundaries (relative to the model's boundaries), thereby ensuring that chunks are not split at points where the language model hallucinates, and avoiding the introduction of more uncertainties when the LLM retrieves and answers questions.
> > >
> > > - Traditional chunking methods treat sentences as independent logical units, whereas we adopt meta-chunks as independent logical units. For instance, in the RAG system, if users opt for a small model and set a relatively low top\_k value for recall, meta-chunks can be directly utilized. However, in cases where users employ LLMs with extended contexts and require larger text chunks, meta-chunks can initially be generated and subsequently merged based on the desired chunk size to achieve the final chunking outcome.
> > >
> > > We sincerely hope that this response clarifies your questions regarding PPL Chunking.

---

### Official Review · Reviewer_bMRY · 2024-11-01

**Soundness:** 2
**Presentation:** 2
**Contribution:** 2
**Rating:** 3
**Confidence:** 4

**Summary:**

This paper focuses on the retrieval stage of RAG, proposing a new chunking strategy to enhance the efficiency of existing methods, such as similarity-based approaches and LumberChunker while outperforming rule-based methods with minimal computational cost. Experimental results demonstrate that Meta-Chunking improves performance in both single-hop and multi-hop question answering more efficiently.

**Strengths:**

- Meta-chunking utilizes the advantages of large language models trained on large-scale corpora and proposes splitting documents based on PPL calculation or segmentation judgment prompting.
- This method only requires calculating the PPL for each sentence or determining if two parts should be segmented, effectively reducing the need for larger models. Instead, smaller models, such as 1.5B or 7B, are sufficient to meet document segmentation needs.
- Experiments conducted across multiple datasets show that PPL chunking not only outperforms similarity-based methods but also reduces time costs.

**Weaknesses:**

- The performance of meta-chunking appears inconsistent. As shown in Table 1, margin sampling chunking methods often fall behind Llama index and similarity-based chunking, while requiring much more time, suggesting that margin sampling may be less effective.
- Although this paper is titled 'efficient text segmentation via logical perception',  the concept of 'logical perception' is unclear. Given that the authors use PPL for document segmentation, it is uncertain how this approach is logic-based.
- The paper only compares three baselines: original, Llama_index, and similarity chunking. However, as noted in line 267, LumberChunker results for other datasets are missing without sufficient explanation. Additionally, other chunking methods, such as proposition [1] and thread [2], introduce new approaches that are neither compared nor referenced in the related work.

[1] Chen, Tong, et al. "Dense x retrieval: What retrieval granularity should we use?." arXiv preprint arXiv:2312.06648 (2023).

[2] An, Kaikai, et al. "Thread: A Logic-Based Data Organization Paradigm for How-To Question Answering with Retrieval Augmented Generation." arXiv preprint arXiv:2406.13372 (2024).

**Questions:**

- The details about the RAG system are missing, such as the number of retrieved chunks. The number of chunks incorporated by the LLM significantly affects performance. How would increasing the number of chunks impact performance compared to the baselines?
- In Lines 200–202, the authors state that "high PPL indicates the cognitive hallucination of LLMs towards the real content, and such portions should not be segmented." However, high PPL typically suggests that the text deviates from the LLM's training corpus, so it is unclear why this would be considered cognitive hallucination. The method only considers PPL differences between consecutive sentences and segments when PPL reaches a margin, however, there are some misunderstandings about the author's claim.
- According to Table 3, the threshold for meta-chunking is consistently set to zero, which implies that segmentation may occur after each sentence. With dynamic combination being a factor, how does this approach differ from simply splitting documents by sentence and concatenating them to the desired length?

---

> ### Author Response · Authors · 2024-11-24
>
> Thank you for your constructive suggestions. We have refactored the methodology and main results sections of the original paper for clearer presentation.
>
> > W1: The performance of meta-chunking appears inconsistent. As shown in Table 1, margin sampling chunking methods often fall behind Llama index and similarity-based chunking, while requiring much more time, suggesting that margin sampling may be less effective.
>
> Our main innovation is the introduction of the PPL Chunking method, which addresses the long runtime and high instruction-following capability requirements of current LLMs chunking methods. This allows LLMs chunking to be better applied in real-world scenarios with massive text. Since LumberChunker is almost unusable for models of 7B or smaller sizes, it is difficult to compare different methods under the same model. Therefore, we first propose Margin Sampling Chunking to address the high instruction-following capability requirements of the current sota method. As a minor improvement, it is included in the baseline to facilitate comparison with PPL Chunking, but its runtime is too long, similar to LumberChunker. PPL Chunking better addresses both of the aforementioned issues.
>
> > W2: Although this paper is titled 'efficient text segmentation via logical perception', the concept of 'logical perception' is unclear. Given that the authors use PPL for document segmentation, it is uncertain how this approach is logic-based.
>
> In this paper, we categorize text chunking primarily into three types: rule-based, semantic similarity-based, and logic-based. Their instances can be observed through Figure 1 and Figure 2, and their specific differences are as follows:
>
>  - Rule-based: This method divides long texts into paragraphs of fixed length or combines sentences into specific text chunks based on specific rules. However, this approach does not consider the text content and is difficult to adapt to dynamic text changes, resulting in partitioning results that may not align with the actual context.
>
>  - Semantic similarity-based: This approach goes beyond relying solely on the surface structure or fixed rules of the text. Instead, it aims to deeply understand the text content and determine segmentation positions by calculating the semantic similarity between sentences. Nevertheless, the relationship between two sentences is not limited to similarity; there may also be progression, transition, and other contexts. As illustrated in Figure 1, two sentences with a progressive relationship may have low similarity, but they are logically connected and should be grouped together. Characterizing such relationships through semantic similarity alone can be challenging, potentially leading to their separation.
>
>  - Logic-based: LLMs exhibit robust logical reasoning capabilities. Whether it involves utilizing prompts to assess the logical relationship between two sentences or computing the perplexity  variations across consecutive sentences, these models' reasoning abilities can be harnessed to capture deeper linguistic logic among sentences. This actually facilitates more effective guidance for text chunking.

---

> > ### Author Response · Authors · 2024-11-24
> >
> > > W3: The paper only compares three baselines: original, Llama_index, and similarity chunking. However, as noted in line 267, LumberChunker results for other datasets are missing without sufficient explanation. Additionally, other chunking methods, such as proposition [1] and thread [2], introduce new approaches that are neither compared nor referenced in the related work.
> >
> >  - Addressing the issue where current LLMs chunking methods, while superior to traditional approaches, demand significant resources and time, we pose a practical question: How can we fully utilize the powerful reasoning capabilities of LLMs while efficiently accomplishing the text chunking task at a reduced cost? Our research focuses on whether models of 7B or even smaller sizes can perform chunking effectively and quickly. Notably, LumberChunker is inapplicable to models of 7B size or smaller. This method exhibits chaotic output on datasets beyond Qasper, making proper chunking impossible. As mentioned in W1, for comparative purposes, we introduced Margin Sampling Chunking as an intermediary step to facilitate a comparison with PPL Chunking.
> >
> >  - Dense x retrieval [1]: This paper introduces a new retrieval granularity called 'proposition', which compresses and segments text through the training of an information extraction model. We have included an additional experiment related to this method in Table 1. Observations indicate that this approach does not exhibit remarkable performance, has a long chunking time, and is designed specifically for English texts.
> >
> >  - Thread [2]: This paper targets 'how-to' questions, extracting relevant information from documents on a specific topic to construct logical units, akin to instruction manuals. The paper concludes by stating that this method is unsuitable for knowledge-intensive RAG tasks, as such tasks often involve numerous topics and flexible associations, making the construction of logical units challenging. Our experiments primarily focus on single-hop and multi-hop knowledge-intensive tasks. Due to the low correlation between the research question addressed in this paper and text segmentation, we did not compare this method.
> >
> > [1] Chen, Tong, et al. "Dense x retrieval: What retrieval granularity should we use?." arXiv preprint arXiv:2312.06648 (2023).
> >
> > [2] An, Kaikai, et al. "Thread: A Logic-Based Data Organization Paradigm for How-To Question Answering with Retrieval Augmented Generation." arXiv preprint arXiv:2406.13372 (2024).

---

> > > ### Author Response · Authors · 2024-11-24
> > >
> > > Thank you for your questions. We hope to address your concerns and doubts by clarifying our experiments and methods.
> > >
> > > > Q1: The details about the RAG system are missing, such as the number of retrieved chunks. The number of chunks incorporated by the LLM significantly affects performance. How would increasing the number of chunks impact performance compared to the baselines?
> > >
> > >  We strictly followed the control variable strategy, maintaining a consistent block length for various blocking methods on each dataset, as detailed in Appendix 2. Experiments conducted on the same dataset only differed in the blocking method. For ease of reference, we also provided some key information in the experimental settings section.
> > >
> > > > Q2: In Lines 200–202, the authors state that "high PPL indicates the cognitive hallucination of LLMs towards the real content, and such portions should not be segmented." However, high PPL typically suggests that the text deviates from the LLM's training corpus, so it is unclear why this would be considered cognitive hallucination. The method only considers PPL differences between consecutive sentences and segments when PPL reaches a margin, however, there are some misunderstandings about the author's claim.
> > >
> > > 1. Firstly, let's revisit the purpose of text chunking, which is primarily to pass text chunks to the question-answering model for content generation. A high perplexity value indicates greater uncertainty within the LLM regarding the given text, or as you mentioned, a deviation from the training data, where the model's generalization performance is weaker. If we perform segmentation at such locations, where the model exhibits uncertainty, it is likely to introduce more uncertain information to subsequent models. This can affect the model's retrieval and answering performance, increasing the likelihood of errors.
> > >
> > > 2. Our PPL Chunking method involves using LLMs to calculate the PPL value for each sentence in an article and then identifying the minimum values of these PPL scores as potential segmentation points. Moreover, when calculating the PPL value for each sentence, the method takes into account the preceding sentences, ensuring contextual coherence. Therefore, when analyzing the PPL distribution, this approach considers the influence of preceding sentences on segmentation points.
> > >
> > >
> > > > Q3: According to Table 3, the threshold for meta-chunking is consistently set to zero, which implies that segmentation may occur after each sentence. With dynamic combination being a factor, how does this approach differ from simply splitting documents by sentence and concatenating them to the desired length?
> > >
> > > Thank you for raising this question. We would like to clarify some misunderstandings about our proposed PPL chunking method.
> > >
> > >  1. As explained in Q2 regarding the PPL Chunking, we identify the minimum points in the PPL distribution as segmentation points. The threshold is set to zero, indicating that these minimum points may have only a slight decrease relative to the data on both sides, but the decrease is strictly greater than 0, as described in Formula (2). Unless every sentence in the text has the same PPL value, it is unlikely that each sentence will be segmented into a separate text chunk.
> > >
> > >  2. Our approach differs from simply splitting the document by sentences and concatenating them to achieve the desired length. In fact, the Llama_index method in the baseline dynamically merges sentences based on chunk length, without considering the logical integrity between sentences. Traditional chunking methods treat sentences as independent logical units, whereas we adopt meta-chunks as independent logical units and dynamically merge them. For instance, in the RAG system, if a user selects a small model and sets a relatively low top-k value for recall, meta-chunks can be directly utilized. However, if the user employs a long-context LLM and requires larger text chunks, meta-chunks can be generated first and then merged based on the desired chunk size to achieve the final chunking result.
> > >
> > >  We hope this explanation clarifies the implementation process of PPL Chunking. Thank you again for your insightful questions.

---

> > > > ### Comment · Reviewer_bMRY · 2024-11-28
> > > >
> > > > Thanks for the authors' responses, which partly address my concerns regarding aspects like baselines and the threshold for PPL chunking.
> > > >
> > > > Although the authors reorganized Table 1, and the new table appears to support their claims, my initial concern remains unresolved. Specifically, margin sampling chunking underperforms compared to the LlamaIndex across different model sizes. Given the authors' statement that "Margin Sampling Chunking serves as an intermediary step to facilitate a comparison with PPL Chunking," Is there any explanation in the revised paper? And if it is expected that margin sampling performs poorly on smaller models?
> > > >
> > > > For Q2, is there any evidence to support the author's point rf point 1?
> > > >
> > > > Additionally, it would be helpful if the authors used different colors to highlight the modified content.
> > > >
> > > > Based on this, I will maintain my initial score.

---

> > > > > ### Author Response · Authors · 2024-11-29
> > > > >
> > > > > Dear reviewer bMRY,
> > > > >
> > > > > We sincerely appreciate your feedback. Regarding the questions you raised, we would like to clarify them accordingly.
> > > > >
> > > > > > Q1: Specifically, margin sampling chunking underperforms compared to the LlamaIndex across different model sizes. Given the authors' statement that "Margin Sampling Chunking serves as an intermediary step to facilitate a comparison with PPL Chunking," Is there any explanation in the revised paper?
> > > > >
> > > > > - We provide explanations in lines 288 to 292 of the paper. In response to the issue that current large model chunking methods outperform traditional ones but require significant resources and time costs, we pose a pertinent question: How can we fully leverage the powerful reasoning capabilities of LLMs while efficiently accomplishing the text chunking task at a reduced cost? Therefore, our research focuses on whether models of 7B or even smaller sizes can perform chunking well and rapidly (lines 78-83).
> > > > >
> > > > > - Our primary innovation lies in proposing the PPL Chunking approach, which addresses the issues of long runtime and high instruction-following capability requirements of current large model methods. Meanwhile, LumberChunker is barely applicable to models of 7B or smaller scales. Due to LumberChunker's limited applicability to models of 7B or smaller, it is challenging to conduct comparisons among different methods using the same model. Therefore, we first introduce Margin Sampling Chunking to solve the high instruction-following capability requirements of this sota method and serve as an intermediate transition for comparison with PPL Chunking. For clearer presentation, we have included Margin Sampling Chunking in the baseline, as shown in lines 293-305 of the paper.
> > > > >
> > > > > > Q2: And if it is expected that margin sampling performs poorly on smaller models?
> > > > >
> > > > > As mentioned in Q1, the Margin Sampling Chunking method is an improvement upon LumberChunker, primarily addressing its inability to be applied to smaller models. As a baseline, it further highlights the challenges of solving chunking problems using small models. In contrast, our PPL Chunking approach achieves performance improvements compared to traditional methods across 0.5B, 1.5B, and 7B models. Meanwhile, compared to current large model chunking methods, it also reduces time and resource consumption, achieving a balance between performance and cost. Currently, most models employed in similarity-based chunking methods are above 0.5B in size. Language models below 0.5B lack sufficient linguistic capabilities, which is why we did not conduct experiments on models smaller than 0.5B.
> > > > >
> > > > > >Q3: For Q2, is there any evidence to support the author's point rf point 1?
> > > > >
> > > > > In the [1], a thorough examination and analysis of the hallucination problem in large language models is conducted, providing deeper insights into model hallucinations, which is cited in line 33 of our paper. Specifically, for the context of text chunking, we have provided a theoretical explanation in the THEORETICAL ANALYSIS OF PPL CHUNKING section, spanning lines 200-258 of our paper.
> > > > >
> > > > > >Q4: Additionally, it would be helpful if the authors used different colors to highlight the modified content.
> > > > >
> > > > > Thank you for your constructive suggestions. We have highlighted the revised sections using yellow marking, but we found that the paper submission system currently does not support further submissions. Therefore, we are annotating the revised parts here for your reference: 16-19, 25-27, 47-52, 87-89, 99-105, 182-185, 283-285, 288-305, 309-313, 348-382.
> > > > >
> > > > > We would appreciate it greatly if you could let us know whether our responses address your concerns and adjust your rating accordingly.
> > > > >
> > > > > Sincerely,
> > > > >
> > > > > Authors of Submission 4310

---

### Official Review · Reviewer_YzGf · 2024-11-02

**Soundness:** 3
**Presentation:** 1
**Contribution:** 2
**Rating:** 5
**Confidence:** 2

**Summary:**

This paper introduces Meta-Chunking, a text segmentation method designed to enhance the chunking phase in Retrieval-Augmented Generation (RAG) systems. The approach operates at a granularity between sentences and paragraphs by capturing logical relationships between sentences. The authors propose two specific strategies: margin-based sampling and perplexity-based chunking, validating their approach through experiments on 11 datasets. The results demonstrate significant improvements over traditional methods in both single-hop and multi-hop QA tasks, while maintaining computational efficiency.

**Strengths:**

1. Introduces the concept of Meta-Chunks, effectively bridging the gap between sentence-level and paragraph-level segmentation
2. Presents two complementary strategies: the margin-based approach reduces model dependency, while the perplexity method improves processing efficiency. The dynamic merging strategy thoughtfully balances fine and coarse granularity
3. Provides comprehensive experimental validation across multiple datasets and evaluation metrics, supported by thorough theoretical analysis and ablation studies

**Weaknesses:**

1. The paper's motivation could be better articulated. The authors seem to overlook the fundamental discussion of why text chunking is necessary in the first place, and don't clearly identify the specific limitations of existing chunking methods. The significance of the chunking phase within the RAG pipeline deserves more thorough examination
2. The logic behind Table 1 is somewhat questionable. Why weren't different methods compared using the same LLM? Additionally, there's an open question about whether the text chunking method might show more significant improvements with smallerLLM while offering smaller improvements to larger LLMs.

**Questions:**

Please refer to the weakness part.

---

> ### Author Response · Authors · 2024-11-24
>
> > W1: The paper's motivation could be better articulated. The authors seem to overlook the fundamental discussion of why text chunking is necessary in the first place, and don't clearly identify the specific limitations of existing chunking methods. The significance of the chunking phase within the RAG pipeline deserves more thorough examination
>
> Thank you for your thoughtful questions. We have made improvements and enhancements based on the original paper.
>
> 1. When introducing the problem, we discussed the necessity of RAG and the limitations of current chunking in the pipeline, but did not address why chunking is needed. To this end, we have added additional explanations in lines 47-52 of the introduction section.
>
>     - By delicately splitting long documents into multiple chunks, this module not only significantly improves the processing efficiency and performance of the system, reducing the consumption of computing resources, but also enhances the accuracy of retrieval [1]. Meanwhile, the chunking strategy allows information to be more concentrated, minimizing the interference of irrelevant information, enabling LLMs to focus more on the specific content of each text chunk and generate more precise responses [2].
>
> 2. Regarding the specific limitations of existing chunking methods, we have made comparisons in multiple places in the paper. By further clarifying, we hope to address your concerns and provide you with a clear understanding of shortcomings of current chunking methods.
>
>     - In lines 52-83 of the introduction section, we highlight the shortcomings of current dynamic chunking methods and provide an example illustration through Figure 1. Currently, semantic similarity chunking cannot deeply capture the logical relationships between sentences, while LLMs chunking methods are slow and computationally expensive, making them difficult to adapt to real-world scenarios with massive text.
>
>     - In lines 156-160 of the methodology section, we compare our method with existing methods while elaborating on our approach, and provide a visual demonstration in Figure 2. As can be seen, current methods either directly split the sentence or fail to capture the logical connections between sentences.
>
>     - Additionally, we have moved the related work section from the end to the beginning to better provide relevant background knowledge.
>
> [1] Maciej Besta, et al. Multi-head rag: Solving multi-aspect problems with llms. arXiv preprint arXiv:2406.05085, 2024.
>
> [2] Weihang Su, et al. Dragin: Dynamic retrieval augmented generation based on the real-time information needs of large language models. arXiv
> preprint arXiv:2403.10081, 2024.
>
> > W2: The logic behind Table 1 is somewhat questionable. Why weren't different methods compared using the same LLM......whether the text chunking method might show more significant improvements with smallerLLM......
>
> 1. Thank you for highlighting this point. Since LumberChunker is almost inapplicable to models of 7B or smaller sizes, and we controlled variables to ensure consistent chunk lengths across different chunking methods for the same dataset, which was difficult for some baselines to adapt to, we were unable to fully demonstrate the results. Therefore, we reconstructed main results and migrated the content about Margin Sampling Chunking from the method section to the baseline section to align with the changes in the main experiment.
>
>     - Our main innovation is the PPL Chunking method, which addresses issues of long runtime and high instruction-following requirements of current LLMs methods. Margin Sampling Chunking solves the high instruction-following capability requirement of the current sota method, but its runtime is still too long. Therefore, it is included as a minor improvement in the baseline for comparison with PPL Chunking.
>
>     - Meanwhile, we explored the performance and time consumption of different methods on three models of different scales: Qwen2-0.5B, Qwen2-1.5B, and Qwen2-7B.
>
> 2. By adding experiments with the 0.5B model, we further explored the question of whether using a smaller model could bring more significant improvements to RAG.
>
>     - We found that on the 2WikiMultihopQA and MultiFieldQA-zh datasets, chunking with the 0.5B model performed better than the 1.5B model. In other cases, the performance decreased slightly, but it was still better than the baseline. This confirms the flexibility of PPL Chunking, indicating that it does not have strict requirements on model performance.
>
>     - Moreover, as the model scale decreases, the execution time for text chunking tasks decreases significantly, reflecting the advantage of small models in improving processing efficiency. This is consistent with our motivation to use small language models to solve the text chunking problem.
>
> We sincerely hope this response clarifies your concerns. If you have any further questions, we would be happy to discuss them in more detail.

---

### Note · Authors · 2024-12-10

I have read and agree with the venue's withdrawal policy on behalf of myself and my co-authors.